# EMBODIEDGS: RECONSTRUCT UNIFIED EMBODIED REPRESENTATION FROM RGB STREAM

## ABSTRACT

This paper addresses the challenge of incrementally reconstructing object-centric 3D representations from only a pose-free RGB video stream. Existing dense SLAM methods face a dual challenge: they are constrained by a reliance on precise camera poses and RGB-D input for initialization, and they lack precise instance-level scene understanding. Moreover, the quality of their reconstruction and perception is fragile to systematic errors. To this end, we propose EmbodiedGS, a pipeline that jointly performs incremental 3D reconstruction and perception from RGB stream to constructs an Object-Centric 3D Gaussians (OCGS) representation that is both geometrically accurate and rich with instance-level information. Specifically, our approach leverages MASt3R-SLAM for Gaussian geometric initialization and introduces a Global-Associated Instance Memory (GAIM) to consistently track objects across views using multi-modal cues. We then construct the initial OCGS by lifting instance information to 3D Gaussians via optimizable binary embeddings. Finally, this representation is refined through a joint optimization process that leverages the synergy between reconstruction and perception to mutually correct inaccuracies, yielding a robust, high-fidelity OCGS. Extensive experiments are conducted on TUM-RGBD and ScanNet datasets and a real-world robotic platform, where EmbodiedGS demonstrates competitive performance even compared with RGB-D SLAM methods and offline 3D instance segmentation methods. Code will be released. Project page.

## 1 INTRODUCTION

Reconstructing 3D scenes from visual observations is a fundamental problem in computer vision. Recently, 3D Gaussian Splatting (Kerbl et al., 2023) has emerged as a powerful paradigm, achieving state-of-the-art rendering quality and efficiency. However, applying 3DGS-based methods to real-world embodied agents (Kwon et al., 2023; Lu et al., 2024; Wu et al., 2024) faces several critical limitations. Many existing approaches require offline optimization, lack vital instance-level information for interaction, and often depend on high-precision RGB-D sensors, limiting their versatility. To facilitate practical deployment, we contend a unified 3D representation should be: (1) **Incremental**. Enables synchronous 3D reconstruction as an agent explores, ensuring real-time scene updates for instantaneous decision making. (2) **Object-Centric**. Supports direct instance-level queries, facilitating both scene understanding and interactive manipulation. (3) **Error-resilient**. Overcomes inevitable errors in initial reconstruction and segmentation to build a high-quality scene representation robustly. We propose Object-Centric Gaussian Splatting (OCGS) as a promising solution to fulfill these requirements. Inheriting the explicit nature of point clouds, OCGS naturally supports incremental updates while maintaining object-centric awareness through per-Gaussian instance labeling. Crucially, OCGS ensures its robustness by jointly leveraging geometric and perceptual cues, correcting initial inaccuracies to forge a high-fidelity final model.

We focus on a practical setting: to incrementally reconstruct OCGS from pose-free RGB stream. This imposes minimal assumptions on the input data, ensuring broad applicability with only commodity-grade monocular RGB sensors. The most relevant task to this setting is 3DGS-based dense SLAM (Keetha et al., 2024; Yan et al., 2024; Matsuki et al., 2024; Zhu et al., 2024a), which tracks camera pose online and simultaneously maintains 3D Gaussian map as scene representation. To empower Gaussian-based SLAM system with scene understanding ability, recent works (Li et al., 2024; Zhu et al., 2024b) propose semantic visual SLAM by incorporating semantic features into 3D

Figure 1: We present EmbodiedGS, a framework for incrementally reconstructing Object-Centric Gaussian Splatting (OCGS) representations from only a monocular, pose-free RGB stream. This task is challenging, as initial perception often suffers from temporal inconsistencies across views, while geometric reconstruction is typically noisy (Left). EmbodiedGS addresses this by introducing a synergistic process where multi-view aggregation refines perception and instance-guided smoothing corrects the geometry (Middle). This mutual refinement yields a unified, high-fidelity OCGS map that is both geometrically accurate and rich with consistent, instance-level semantics (Right).

Gaussians to support semantic rendering and category-wise scene editing. However, most existing 3D Gaussian SLAM approaches rely on RGB-D sensor or large monocular depth estimation model for Gaussian initialization. Their scene understanding capability is also limited to category-level segmentation, and exhibits a strong dependency on existing 2D models and a high sensitivity to segmentation errors. Therefore, achieving efficient dense 3D Gaussian reconstruction with instance-level understanding from monocular RGB streams remains an open challenge.

In this paper, we propose EmbodiedGS, an efficient pipeline to incrementally reconstruct OCGS scene representations from unposed RGB streams. Different from a naive fusion of dense 3D Gaussian SLAM and 3D instance segmentation, our approach establishes a synergistic loop between reconstruction and perception. It fully exploits the 3D Gaussian representation for incremental instance understanding, which reciprocally aids in the robust optimization of the OCGS map. Specifically, buiding upon MASt3R-SLAM (Murai et al., 2024), which efficiently estimates camera poses and generates coordinate-aligned pointmaps from an unposed RGB stream, we implement a submap-wise OCGS mapping system. This system begins by initializing the geometry of 3D Gaussians with the output of MASt3R-SLAM. Concurrently, to incorporate object-level information into the geometry, we build a Global-Associated Instance Memory (GAIM) by incrementally associating instance proposals across views with multi-modal matching criteria to ensure global consistency. The resulting instance labels from GAIM are then assigned to the Gaussians, forming an initial OCGS. Finally, to address inevitable inaccuracies from initial reconstruction and segmentation, we jointly optimize the OCGS by leveraging the synergy between perception and reconstruction, simultaneously refining Gaussian geometry and instance embedding to produce a high-fidelity representation. We conduct extensive experiments on the popular TUM-RGBD and ScanNet datasets. Our EmbodiedGS achieves appealing performance in both instance segmentation and rendering. Furthermore, we deploy EmbodiedGS on a real-world robot, showcasing its remarkable capability to reconstruct complex scenes while effectively handling moving objects, confirming its practical applicability.

## 2 RELATED WORK

**3D Reconstruction from Video Stream.** Methods for reconstructing 3D geometry from video streams primarily fall into two categories: dense visual SLAM and end-to-end approaches. Dense visual SLAM has evolved from classical point-cloud systems (Engel et al., 2014; Tateno et al., 2017; Newcombe et al., 2011; Zhou et al., 2018) to recent methods leveraging 3D Gaussian Splatting (Kerbl et al., 2023) for real-time rendering (Huang et al., 2024; Keetha et al., 2024; Matsuki et al., 2024; Zhu et al., 2024a). However, these Gaussian-based methods often rely on RGB-D input for Gaussian initialization and require costly test-time optimization, which severely limits camera tracking speed. In contrast, end-to end methods like Spann3R (Wang & Agapito, 2024) and CUT3R (Wang et al., 2025b) enhance DUSt3R (Wang et al., 2024) with spatial memory and continuous updating states respectively, enabling real-time, incremental point cloud reconstruction from only pose-free RGB streams. While efficient, end-to-end methods often struggle with catastrophic forgetting on long sequences. Recently, MASt3R-SLAM (Murai et al., 2024) bridged this gap by

integrating the local end-to-end reconstruction capabilities of MASt3R (Leroy et al., 2024) within a global SLAM framework, achieving real-time,high-quality point cloud reconstruction. Building on this, our EmbodiedGS extends MASt3R-SLAM by introducing an object-centric 3D Gaussian representation, leveraging perceptual cues and Gaussian optimization mechanism to mitigate point cloud prediction errors for more robust performance.

**Incremental 3D Scene Understanding.** To enable embodied agents to understand unknown environments on-the-fly, incremental (or online) 3D scene understanding emerges as a pivotal yet unresolved research challenge. Early methods (McCormac et al., 2017; Narita et al., 2019) typically projected per-frame 2D predictions onto a 3D point cloud for fusion, a step often compromised by a lack of geometric and temporal awareness. To address this, Fusion-aware 3D-Conv (Zhang et al., 2020) and SVCNN (Huang et al., 2021) construct data structures to maintain the information of previous frames and conduct point-based 3D aggregation to fuse the 3D features for semantic segmentation. INS-CONV (Liu et al., 2022) extends sparse convolution (Graham et al., 2018; Choy et al., 2019) to incremental CNN to efficiently extract global 3D features for semantic and instance segmentation. In order to simplify the model design, Online3D (Xu et al., 2024) proposes a new paradigm that empowers offline model with online perception ability by multimodal memory-based adapters. EmbodiedSAM (Xu et al., 2025b) further leverages 2D VFM to achieve real-time and fine-grained 3D instance segmentation. However, these methods are performed on posed RGB-D videos. Both camera pose and depth sensor are required to form 3D geometry of the scene. Differently, our EmbodiedGS jointly performs incremental 3D reconstruction and 3D perception, which fully exploits the correlation between geometry and instance information to build a robust system.

## 3 APPROACH

Given a streaming RGB video $\mathcal{V}_t = \{I_1, I_2, ..., I_t\}$ with known camera intrinsics $K$, our system aims to simultaneously estimate camera trajectory, reconstruct 3D Gaussian representation of the scene, and predict globally consistent 3D instance labels on the reconstruction, thereby constructing an OCGS representation of the scene. All tasks should be performed online as the video stream progresses. At any time instant $t$, future images $I_k$ $(k > t)$ are not available.

**Overview.** The overall pipeline of our approach is illustrated in Figure 2. We first initialize the geometry of 3D Gaussians using camera poses and pointmaps predicted by MASt3R-SLAM from monocular RGB stream, while concurrently building a Global-Associated Instance Memory (GAIM) by incrementally associating instance proposals across views using multi-modal matching criteria for global consistency. The resulting globally consistent instance information from GAIM is then leveraged to assign instance labels to the Gaussians, forming the initial Object-Centric 3D Gaussians (OCGS). Finally, to correct for initial reconstruction and segmentation inaccuracies, we perform a joint optimization that leverages the synergy between perception and reconstruction to mutually refine the multi-view consistency of GAIM and the OCGS scene representation.

### 3.1 PRELIMINARY

Our system is built upon two preliminary modules for geometry and perception initialization.

**For geometric reconstruction and camera tracking,** we utilize MASt3R-SLAM (Murai et al., 2024) to process the monocular RGB stream input, which provides the initial geometry for our 3D Gaussians. As a keyframe-based system, it uses the two-view reconstruction model MASt3R (Leroy et al., 2024) to process the current frame $I_t$ and last keyframe $I_k$, obtaining pointmaps $\mathbf{X}_t^k, \mathbf{X}_t^t \in \mathbb{R}^{H \times W \times 3}$ with their confidences $\mathbf{C}_t^k, \mathbf{C}_t^t \in \mathbb{R}^{H \times W \times 1}$, and $d$-dimensional matching features $\mathbf{D}_t^k, \mathbf{D}_t^t \in \mathbb{R}^{H \times W \times d}$ with the corresponding confidences $\mathbf{Q}_t^k, \mathbf{Q}_t^t \in \mathbb{R}^{H \times W \times 1}$. Here we use notation $\mathbf{X}_t^k$ to represent the pointmap of image $I_k$ expressed in the coordinate frame of $I_t$. Based on these outputs and the pre-stored keyframe pointmap $\mathbf{X}_k^k$ of $I_k$, the SLAM system establishes point correspondences between the two frames to solve the relative pose transformation $\mathbf{T}_{kt} \in \mathbf{Sim}(3)$. It also includes standard loop closure and backend optimization to ensure global pose consistency.

**For 2D instance perception,** we employ the efficient open-vocabulary segmenter YOLO-E (Wang et al., 2025a) to generate a set of instance masks $\{M_m\}_{m=1}^L$ for current frame $I_t$. To better harness the information inherent in the masks, we extract both geometric and visual descriptors for each

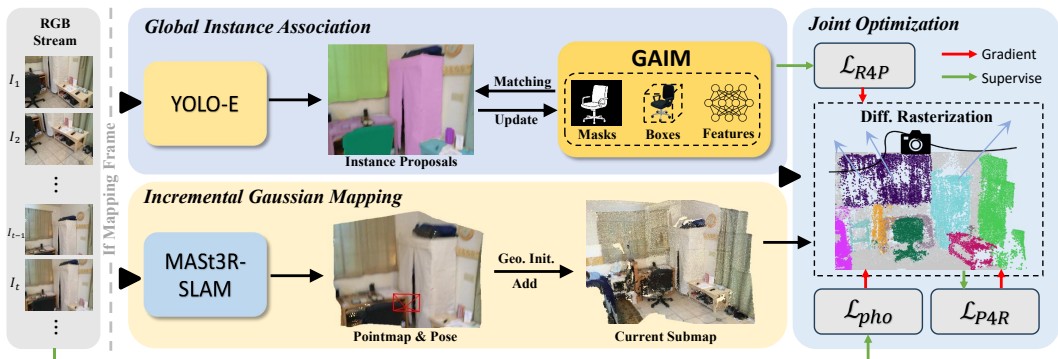

Figure 2: Framework of *EmbodiedGS*. Our system receives pose-free RGB stream and incrementally builds object-centric 3D Gaussians (OCGS). It features two parallel branches: *Global Instance Association* builds a consistent instance memory (GAIM) across views, while *Incremental Gaussian Mapping* uses MASt3R-SLAM to construct the 3D geometry of Gaussians. Both outputs are fused and refined in a *Joint Optimization* stage to enhance perception and reconstruction quality.

instance. On one hand, we leverage the pixel-wise correspondence between the pointmap $\mathbf{X}_t^t$ and the image $I_t$ to extract the point cloud of each instance with its mask $M_m$ and compute its 3D bounding boxes $B_m$ as the geometric descriptor. On the other hand, inspired by Duisterhof et al. (2024), we reuse the highly discriminative feature map of $I_t$ from the MASt3R encoder, applying average pooling within each mask region to obtain a feature vector $F_m$ as the visual descriptor. Finally, we combine these descriptors into a set of instance proposals for the current frame, denoted as $P_t = \{p_m\}_{m=1}^L$, where each proposal is a triplet $p_m = (M_m, B_m, F_m)$. These proposals are then fed into our GAIM module for global association and memory updates.

## 3.2 OCGS Initialization with GAIM Update

**Submap-based Gaussian Mapping.** To manage computational and memory costs, we employ a submap-based mapping approach. The entire scene is partitioned into sequential submaps, with each keyframe selection from MASt3R-SLAM finalizing the current submap and initializing a new one, thus ensuring efficient resource management. Within each submap, we further select a sparse subset of distinctive frames for mapping, termed mapping frames, which serve a dual purpose. Geometrically, their pointmaps are used to initialize new 3D Gaussians, and their RGB images supervise the Gaussian optimization process. Semantically, their corresponding instance proposals are associated with GAIM to establish globally consistent instance IDs, which are then lifted to the newly created 3D Gaussians to form the initial OCGS representation.

When $I_t$ is a mapping frame, we transform its pointmap $\mathbf{X}_t^t$ into the coordinate frame of $I_k$ with relative pose $\mathbf{T}_{kt}$ and initialize 3D Gaussians from it, denoted as $\mathcal{G}_t^{cur}$. Each 3D Gaussian is represented as $\mathbf{g} = (\boldsymbol{\mu}, \mathbf{s}, \mathbf{r}, \alpha, \mathbf{c})$, where $\boldsymbol{\mu}$ is centroid, $\mathbf{s}$ is scaling vector, $\mathbf{r}$ is rotational quaternion, $\alpha$ is opacity value and $\mathbf{c}$ is color. Here $\boldsymbol{\mu}$ and $\mathbf{c}$ are initialized with the transformed pointmap. $\mathcal{G}_t^{cur}$ will be appended to $\mathcal{G}_{t-1}^{sub}$, the previous reconstructed 3D Gaussians in current submap, to obtain $\mathcal{G}_t^{sub}$. We denote mapping frames selected so far within current submap as $\mathcal{V}_t^{map} = \{I_1^{map}, I_2^{map}, ..., I_m^{map}\}$. We optimize the geometry and appearance of $\mathcal{G}_t^{sub}$ via differentiable rendering on $\mathcal{V}_t^{map}$ using photometric loss, which is a weighted sum of an $\mathcal{L}_1$ and an SSIM (Wang et al., 2004) term:

$$\mathcal{L}_{pho} = (1 - \lambda_{SSIM}) \cdot |I_m^{map} - \hat{I}_m^{map}|_1 + \lambda_{SSIM} \cdot (1 - \text{SSIM}(I_m^{map}, \hat{I}_m^{map})), \qquad (1)$$

where $\hat{I}_m^{map}$ denotes the image rendered by $\mathcal{G}_t^{sub}$. To constrain the excessive elongation of Gaussians in sparsely observed regions, we introduce an isotropic regularization term:

$$\mathcal{L}_{reg} = \frac{1}{K} \sum_{k=1}^{K} |s_k - \bar{s}_k|_1, \qquad (2)$$

where $k$ is the number of Gaussians in $\mathcal{G}_t^{sub}$, $s \in \mathbb{R}^3$ is the scales of a Gaussian and $\bar{s}$ is its mean.

**GAIM Update via Multi-Cue Matching.** To convert the initial Gaussian mapping into OCGS, each 3D Gaussian must be assigned a globally consistent instance ID. The core challenge lies in associating new instance proposals in current frame with instances observed previously to maintain consistency. We address this with our proposed Globally-Associated Instance Memory (GAIM), a dynamic memory bank that stores and incrementally updates the attributes of previously seen instances, $O = \{o_1, o_2, ..., o_N\}$. Each instance entry $o_i$ comprises multi-view 2D masks, an aggregated visual descriptor, and a 3D bounding box.

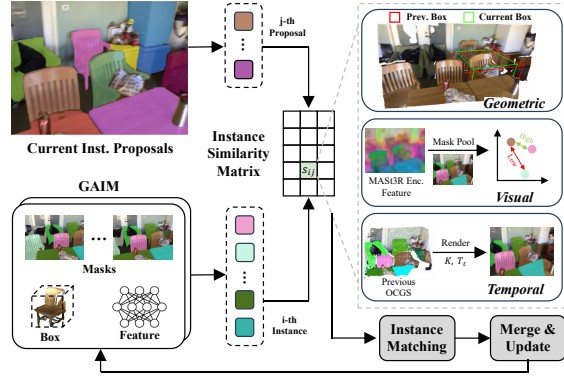

Figure 3: Illustration of GAIM update. We design three merging criteria to compute the instance similarity matrix between current instance proposals and global instances, which achieves accurate instance matching and supports efficient GPU parallelization.

When a new set of $L$ instance proposals $P_m = \{p_1, p_2, ..., p_L\}$ is acquired from current mapping frame $I_m^{map}$, we update GAIM by solving a bipartite matching problem between these proposals and existing instances $O$. The matching is determined by a similarity matrix $S = [s_{ij}]$, where $s_{ij}$ is the similarity score between $o_i$ and $p_j$. For each successfully matched pair, the attributes of proposal from $P_m$ are merged into its corresponding instance in $O$. Proposals that remain unmatched are considered new objects and registered with new global IDs in the memory $O$. To ensure accurate and robust matching, we design three merging criteria for similarity computation, considering *geometric*, *visual* and *temporal* cues between each pair, as shwon in Figure 3.

*Geometric similarity* measures spatial proximity to encourage merging of nearby instances and prune incorrect merges between distant instances. We obtain this score $s_{ij}^{geo}$ by computing the 3D box DIoU(Zheng et al., 2020) between the stored instance box $o_i.\texttt{box}$ and the proposal box $B_j$, which considers both the spatial overlap and the center-to-center distance between them. We opt for 3D boxes rather than 3D masks as they are more efficient to store, update and compare. We have:

$$s_{ij}^{geo} = \text{DIoU}(o_i.\texttt{box}, B_j), \tag{3}$$

If a merge ocurrs, $o_i.\texttt{box}$ is updated to be the minimal bounding box enclosing the two boxes.

*Visual similarity* captures appearance resemblance between instances to distinguish semantically different objects. We measure this score $s_{ij}^{vis}$ using the cosine similarity between the stored visual descriptor $o_i.\texttt{vis}$ and the visual descriptor $F_j$ of proposal $p_j$:

$$s_{ij}^{vis} = \text{Cos}(o_i.\texttt{vis}, F_j), \tag{4}$$

Once merged, $o_i.\texttt{vis}$ is updated using a running average: $o_i.\texttt{vis} \leftarrow \frac{n \cdot o_i.\texttt{vis} + F_j}{n+1}$, where $n$ is the numbers of previous merges.

*Temporal similarity* leverages spatio-temporal consistency cue to track an object across frames. Instead of using an external video segmentation model as instance propagator, we directly leverage our previously reconstructed map $\mathcal{G}_{t-1}^{sub}$ to render the mask of an existing object $o_i$ using the estimated pose $\mathbf{T}_t$. Then the temporal similarity $s_{ij}^{tem}$ can be calculated as the co-coverage between this pre-rendered mask and the proposal's mask $M_j$, where co-coverage between mask pairs is defined as: $\text{CoCov}(M_i, M_j) = \frac{1}{2}(\frac{|M_i \cap M_j|}{|M_i|} + \frac{|M_i \cap M_j|}{|M_j|})$. So we have:

$$s_{ij}^{tem} = \text{CoCov}([\mathcal{R}(\mathcal{G}_{t-1}^{sub}, \mathbf{T}_t)]_i, M_j). \tag{5}$$

Here $\mathcal{R}$ is mask rendering operation (detailed in Sec. 3.3) and $[\cdot]_i$ selects the mask for instance $o_i$. This approach is highly efficient, as all instance masks can be generated in a single rendering pass.

Note that these three scores can be computed in parallel, enabling efficient matrix operations. We aggregate them via a weighted sum, $s_{ij} = \lambda_1 s_{ij}^{geo} + \lambda_2 s_{ij}^{vis} + \lambda_3 s_{ij}^{tem}$, to form the final similarity matrix $S$. Then we solve the bipartite matching problem on $S$, discarding matches where $s_{ij} < \sigma_m$. Based on the resulting associations, each new proposal is either merged into an existing instance in GAIM or registered as a new one. Finally, these newly assigned global instance IDs are lifted from 2D proposals to their corresponding 3D Gaussians in $\mathcal{G}_t^{cur}$, completing the OCGS initialization.

### 3.3 OCGS Refinement via Joint Optimization

The initial OCGS representation constructed in Sec. 3.2 is prone to two primary error sources, which leads to degraded mapping quality. The first is reconstruction error, arising from inaccurate pointmap depths and pose drift from MASt3R-SLAM. The second is perception error, such as missed or over-segmented instances from the 2D segmenter. To mitigate these issues, we first enhance segmentation robustness by introducing a binary instance embedding, which can be gradually optimized. Furthermore, we leverage multi-view perception results to jointly optimize the OCGS, ultimately improving the accuracy of both the reconstruction and perception.

**Explicit Binary Instance Embedding.** Fixed instance IDs are sensitive to segmentation errors. To address this, we attach each Gaussian with an optimizable instance embedding $\mathbf{e} \in \mathbb{R}^d$ to enhance robustness, such that $\mathbf{g} = (\boldsymbol{\mu}, \mathbf{s}, \mathbf{r}, \alpha, \mathbf{c}, \mathbf{e})$. To ensure global consistency of instance embeddings, we employ explicit binary encoding for instance IDs by converting each decimal ID into its corresponding binary representation, thereby yielding a $d$-dimension vector where $d = \lceil \log_2(N_{max}) \rceil$. $N_{max}$ is the pre-defined maximum number of instances which is large enough.

For newly added 3D Gaussians $\mathcal{G}_t^{cur}$ contributed by current mapping frame $I_m^{map}$, we initialize their $d$-dimension instance embedding $\mathbf{e}_i$ to the binary encoding of their instance IDs. Then we can splat $\mathcal{G}_t^{sub}$ according to the camera parameters of images in $\mathcal{V}_t^{map}$ to render instance embedding maps:

$$E(p) = \sum_{i \in \mathcal{N}} \mathbf{e}_i f_i(p) \prod_{j=1}^{i-1} (1 - f_j(p)), \tag{6}$$

where $p$ is a pixel, $f_i(p)$ is the influence factor of each Gaussian on that pixel. The explicit nature of our binary encoding allows for straightforward recovery of discrete instance IDs from their embeddings via rounding and base conversion. This enables the transformation of instance embedding maps into instance masks and the classification of Gaussians by their respective instance IDs.

**Bidirectional Joint Optimization.** We propose a virtuous cycle of bidirectional optimization that jointly refines reconstruction and perception. We formulate this as two reciprocal processes: *"Reconstruction for Perception (R4P)"* and *"Perception for Reconstruction (P4R)"*, which work in tandem to progressively correct each other's errors and ultimately enhance the OCGS mapping quality.

On one hand, reconstruction aids perception by providing a unified 3D space via OCGS, where multi-view perceptual results can be fused. This 3D fusion effectively resolves cross-view inconsistencies and improves perceptual accuracy. Specifically, we use associated instance masks $\{\tilde{M}_m\}$ from GAIM to supervise the optimization of instance embeddings via binary cross-entropy loss:

$$\mathcal{L}_{R4P} = \sum_p w(p) \cdot \text{BCE}\left(\sigma(E_m(p)), b(\tilde{M}_m(p))\right), \tag{7}$$

where $\sigma$ is sigmoid function, $b(\cdot)$ maps a mask to its $d$-bit binary encoding. The per-pixel weight $w(p)$ is derived by the YOLO-E confidence for foreground pixels and set to a constant $w_{bg}$ for the background. This method simplifies instance embedding optimization by decomposing it into independent binary classifications, enabling a confidence-weighted fusion of multi-view segmentations.

On the other hand, perception aids reconstruction by leveraging instance information to enforce spatial smoothness, thus reducing severe depth reconstruction errors. Specifically, we introduce a smoothness loss to encourage Gaussians belonging to the same instance to be spatially compact:

$$\mathcal{L}_{P4R} = \sum_n \left( \frac{1}{|\mathcal{G}_n|} \sum_{\mathbf{g}_i \in \mathcal{G}_n} \left( \frac{C_{max}^n}{C_i} ||\boldsymbol{\mu}_i - \bar{\boldsymbol{\mu}}_{\mathcal{N}(i,n)}||^2 \right) \right). \tag{8}$$

For each instance $n$, the loss pulls a Gaussian's center $\boldsymbol{\mu}_i$ towards a robust local target $\bar{\boldsymbol{\mu}}_{\mathcal{N}(i,n)}$, computed as the confidence-weighted mean of its $k$-nearest intra-instance neighbors. The strength of this pull is inversely weighted by the Gaussian's own confidences $C_i$ from MASt3R. This dual-weighting scheme compels lower confidence Gaussians to align with their high-confidence neighbors, reducing intra-object spatial variance to yield smoother reconstructions.

Finally, the overall loss function for the iterative optimization of our OCGS is defined as:

$$\mathcal{L} = \mathcal{L}_{pho} + \mathcal{L}_{reg} + \lambda_{R4P}\mathcal{L}_{R4P} + \lambda_{P4R}\mathcal{L}_{P4R}. \tag{9}$$

For clarity, the pseudocode for the entire algorithm is provided in the Appendix as Algorithm 1.

## 4 EXPERIMENT

In this section, we present a comprehensive evaluation of EmbodiedGS. We first detail the experimental setup, and then compare EmbodiedGS with SOTA baselines on tasks of 3D instance segmentation and Gaussian rendering. We follow this with in-depth ablation studies and efficiency analyses. Finally, we apply our method to real-world scenarios to demonstrate its effectiveness and robustness.

### 4.1 BENCHMARKS AND IMPLEMENTATION DETAILS

**Benchmarks.** We evaluate on two real-world datasets: TUM-RGBD (Sturm et al., 2012) and ScanNet (Dai et al., 2017) datasets. Following previous protocols (Yugay et al., 2023), we selected 3 sequences from TUM-RGBD and 6 sequences from ScanNet, respectively.

**Compared Methods.** For 3D instance segmentation, we compare with zero-shot 3D scene segmentation methods Yang et al. (2023); Xu et al. (2025a) following the setting of Xu et al. (2025b). For training view rendering, we compare with neural rendering-based SLAM methods Zhu et al. (2022); Yang et al. (2022); Mahdi Johari et al. (2022); Liu et al. (2019); Liso et al. (2024); Keetha et al. (2024); Zhu et al. (2024a); Matsuki et al. (2024); Zhang et al. (2024), on TUM-RGBD and ScanNet following the setting of Zhu et al. (2024a).

**Implementation Details.** Frames are selected as mapping frames under either of two conditions: being a keyframe or covering substantially unobserved regions. For hyperparameters, we set $\lambda_1 = \lambda_2 = \lambda_3 = 0.333$, $\lambda_{SSIM} = 0.2$, $\lambda_{P4R} = 1$, $\lambda_{R4P} = 0.1$, $\sigma_m = 0.15$, $w_{bg} = 0.01$. our system runs on a single NVIDIA GeForce RTX 4090 GPU. See Appendix A.4 for more details.

### 4.2 3D INSTANCE SEGMENTATION

Existing offline 3D instance segmentation models rely on pre-reconstructed point clouds, typically requiring ground-truth (GT) depths and poses as input. In contrast, our approach is online and requires only monocular RGB. To ensure a fair comparison, we re-evaluated SAM3D and SAM-Pro3D by replacing GT inputs with the depth and pose predictions from MASt3R-SLAM. According to Table 1, our EmbodiedGS significantly outperforms both RGB and even RGB-D versions of the baselines, whose performance degrades notably when provided with lower-quality pose and depth. Remarkably, our approach even surpasses the original offline models while requiring only RGB inputs. Visualizations of the segmentation results on ScanNet are provided in Figure 6.

### 4.3 GAUSSIAN RENDERING

We evaluate our rendering performance on training views in Table 2. As a submap-based method, we follow the settings of LoopSplat (Zhu et al., 2024a) by aggregating all submaps into a global map using the predicted camera poses and optimizing it for several iterations after processing the sequence for a fair comparison. Our method outperforms all baselines across all datasets in terms of average PSNR, SSIM, and LPIPS, even surpassing RGB-D SLAM methods. Per-scene rendering results are provided in the appendix as Table 6 and Table 7, with some visualizations in Figure 7.

### 4.4 FURTHER ANALYSIS

**Runtime and Memory Usage.** We analyze the efficiency of EmbodiedGS on an RTX 4090 GPU in Table 3. We report the per-frame tracking and mapping runtime of SLAM methods, obtained by

Table 1: **Class-agnostic 3D instance segmentation** results of zero-shot methods on ScanNet. †: w/o GT pose; ‡: w/o both GT pose and depth.

| Method | Metric | 0000 | 0059 | 0106 | 0169 | 0181 | 0207 | Avg. |
|---|---|---|---|---|---|---|---|---|
| *Posed RGB-D* | | | | | | | | |
| SAMPro3D | AP | 8.0 | 15.6 | 8.9 | 5.8 | 14.4 | 16.5 | 11.2 |
| | $AP_{50}$ | 14.7 | 40.7 | 28.0 | 19.4 | 29.8 | 23.1 | 25.4 |
| | $AP_{25}$ | 33.3 | 72.3 | 64.4 | 57.8 | 48.2 | 45.0 | 52.5 |
| SAM3D | AP | 34.7 | 21.5 | 13.4 | 16.6 | 30.1 | 15.5 | 21.0 |
| | $AP_{50}$ | 53.1 | 34.5 | 38.0 | 37.6 | 41.9 | 26.1 | 37.5 |
| | $AP_{25}$ | 69.8 | 71.6 | 68.1 | 66.4 | 60.2 | 58.3 | 65.4 |
| *RGB-D* | | | | | | | | |
| SAMPro3D† | AP | 9.3 | 11.2 | 0.0 | 8.7 | 8.5 | 4.1 | 7.1 |
| | $AP_{50}$ | 20.2 | 28.5 | 2.3 | 16.8 | 15.9 | 7.5 | 15.2 |
| | $AP_{25}$ | 44.2 | 53.1 | 22.0 | 52.6 | 43.0 | 38.1 | 42.5 |
| SAM3D† | AP | 37.9 | 13.5 | 8.0 | 12.2 | 7.0 | 9.2 | 13.9 |
| | $AP_{50}$ | 51.9 | 28.3 | 17.6 | 27.5 | 14.5 | 18.1 | 25.7 |
| | $AP_{25}$ | 67.7 | 59.5 | 62.0 | 60.4 | 53.8 | 48.9 | 58.2 |
| *RGB* | | | | | | | | |
| SAMPro3D‡ | AP | 2.5 | 5.5 | 1.0 | 4.6 | 2.8 | 1.8 | 3.2 |
| | $AP_{50}$ | 8.3 | 12.5 | 4.6 | 10.6 | 7.1 | 3.6 | 7.9 |
| | $AP_{25}$ | 21.5 | 48.5 | 19.0 | 32.8 | 36.5 | 20.6 | 29.6 |
| SAM3D‡ | AP | 15.2 | 8.5 | 4.9 | 13.5 | 5.1 | 1.2 | 7.9 |
| | $AP_{50}$ | 36.3 | 13.4 | 11.9 | 28.5 | 15.2 | 3.5 | 17.2 |
| | $AP_{25}$ | 57.5 | 43.0 | 37.2 | 55.9 | 47.4 | 36.1 | 45.6 |
| **EmbodiedGS** | AP | **32.0** | **22.9** | **23.0** | **31.2** | **14.7** | **38.1** | **26.7** |
| | $AP_{50}$ | **56.4** | **53.0** | **52.5** | **58.2** | **29.7** | **67.8** | **52.4** |
| | $AP_{25}$ | **66.3** | **76.0** | **91.3** | **81.6** | **65.6** | **84.2** | **76.6** |

dividing the total optimization time by the sequence length. Additionally, we provide the per-frame segmentation time for online 3D instance segmentation methods, calculated as the average processing time across all segmented frames. The embedding size shows both the peak submap memory usage during processing and the size of the final optimized Gaussian map. Since our EmbodiedGS employs an adaptive mapping frame selection strategy and a submap-based mapping approach, it achieves high mapping speed while preventing the embedding size from becoming too large during processing. Additionally, the efficient 2D segmenter and association strategy we use also contribute to high segmentation efficiency, which is $20\times$ faster than SAM3D.

**Ablation Studies.** We conduct comprehensive ablation studies to validate the key components of our method design.

First, we ablate our multi-cue matching criteria, reporting the average performance on 6 scenes of the ScanNet dataset. As reported in Table 4, removing any of the three cues degrades segmentation performance, confirming all are essential. Among them, geometric similarity plays the most pivotal role. This aligns with the observation that geometric information provides robust global discriminative capability for objects. Meanwhile, temporal information serves as a crucial complement, ensuring temporal consistency of segmentation across adjacent frames.

Table 2: **Rendering Performance on 2 Datasets**. EmbodiedGS achieves competitive results on real-world datasets, even outperforming RGB-D SLAM methods.

| Dataset | | *TUM-RGBD* | | | *ScanNet* | |
|---|---|---|---|---|---|---|
| Method | PSNR ↑ | SSIM ↑ | LPIPS ↓ | PSNR ↑ | SSIM ↑ | LPIPS ↓ |
| NICE-SLAM | 14.86 | 0.614 | 0.441 | 17.54 | 0.621 | 0.548 |
| Vox-Fusion | 16.46 | 0.677 | 0.471 | 18.17 | 0.673 | 0.504 |
| ESLAM | 15.26 | 0.478 | 0.569 | 15.29 | 0.658 | 0.488 |
| Point-SLAM | 16.62 | 0.696 | 0.526 | 19.82 | 0.751 | 0.514 |
| Loopy-SLAM | 12.94 | 0.489 | 0.645 | 15.23 | 0.629 | 0.671 |
| SplaTAM | 22.80 | 0.893 | 0.178 | 19.14 | 0.716 | 0.358 |
| LoopSplat | 22.72 | 0.873 | 0.259 | 24.92 | 0.845 | 0.425 |
| MonoGS | 18.82 | 0.740 | 0.327 | 18.79 | 0.707 | 0.585 |
| GLORIE-SLAM | 22.36 | 0.890 | 0.240 | 22.45 | 0.843 | 0.355 |
| **EmbodiedGS** | **24.47** | **0.908** | **0.162** | **26.48** | **0.913** | **0.205** |

Next, we validate our bidirectional joint optimization module. The results in Table 5 demonstrate its effectiveness in mutually refining both perception and reconstruction. Specifically, removing the R4P loss causes a drop of approximately 1.0 across all AP metrics, confirming its role in improving segmentation. Besides, ablating the P4R loss reduces rendering quality by 0.86 dB PSNR, validating its contribution to geometric accuracy.

For further discussion on our method's scalability, generalization and potential limitations, please see Appendix A.5.

Table 3: Runtime and Memory Usage on ScanNet `0000`.

| Method | Tracking /Frame(s) ↓ | Mapping /Frame(s) ↓ | Segmentation /Frame(s) ↓ | Embedding Size(MiB)↓ |
|---|---|---|---|---|
| SplaTAM | 2.08 | 0.69 | – | – / 144.40 |
| LoopSplat | 3.24 | 0.79 | – | **5.21** / 93.98 |
| SAM3D | – | – | 1.74 | – / – |
| **Ours** | **0.06** | **0.09** | **0.07** | 15.51 / **50.31** |

Table 4: Effects of the Merging Criteria.

| Method | AP |
|---|---|
| Remove geometric similarity | 18.6 |
| Remove visual similarity | 20.1 |
| Remove temporal similarity | 24.0 |
| **The final model** | **26.7** |

Table 5: Effects of the Bidirectional Optimization.

| Method | PSNR↑ | SSIM↑ | LPIPS↓ | AP | $AP_{50}$ | $AP_{25}$ |
|---|---|---|---|---|---|---|
| Without Ins. Emb. | 26.39 | 0.913 | 0.205 | 24.3 | 49.0 | 73.6 |
| Without R4P | 26.47 | 0.910 | 0.210 | 25.6 | 51.2 | 75.6 |
| Without P4R | 25.62 | 0.904 | 0.218 | 26.5 | 51.4 | 76.0 |
| **The final model** | **26.48** | **0.913** | **0.205** | **26.7** | **52.4** | **76.6** |

## 4.5 REAL-WORLD APPLICATION

We test EmbodiedGS in real-world scenarios to demonstrate its practical performance and adaptability to dynamic scenes where objects are changed over time.

**Online Visualization.** Figure 4 demonstrate the online reconstruction and segmentation process of EmbodiedGS in both indoor and outdoor (Sun et al., 2020) scenes . It showcases our model's ability to build a complete OCGS representation incrementally, performing scene reconstruction, instance segmentation, and global matching simultaneously from only a monocular RGB stream.

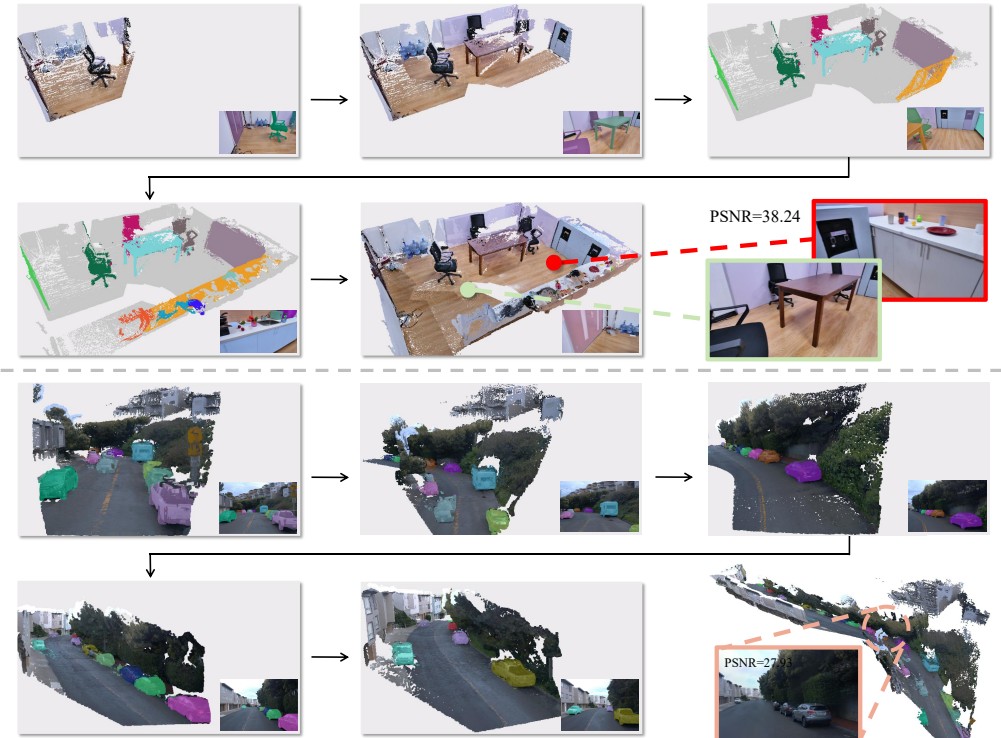

Figure 4: Online visualization of EmbodiedGS on real-world and outdoor scenarios.

**Dynamic Object Handling.** We also demonstrate the adaptability of our OCGS reconstructed in dynamic scenes where objects may move. The system handles changes by performing a consistency check between the map's predictions and current observations, illustrated in Figure 5. For object removal (case 1), it compares the instance mask rendered from the OCGS with the mask segmented from the current frame; if a rendered instance is not observed, its corresponding Gaussians are pruned. For object displacement (case 2), it is treated as a removal from the original location, followed by a re-introduction at the new position by mapping the current observation.

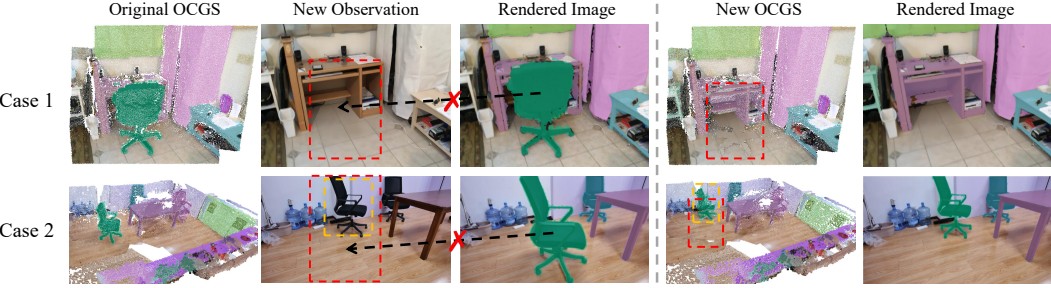

Figure 5: Visualization of object removal and displacement.

## 5 CONCLUSION

In this work, we presented EmbodiedGS, an efficient pipeline for online OCGS reconstruction from RGB video streams. Our approach initializes the 3D Gaussian geometry using MASt3R-SLAM, and concurrently builds a Global-Associated Instance Memory to associate 2D proposals across frames via multi-modal cues, forming the initial OCGS. This initial representation is then refined through a joint optimization process that leverages the synergy between perception and reconstruction, mutually enhancing both the Gaussian geometry and explicit instance embeddings to yield the final OCGS. Extensive experiments on real-world datasets demonstrate EmbodiedGS effectively constructs high-fidelity OCGS, achieving superior performance in both segmentation and rendering quality. We believe the proposed EmbodiedGS provides a unified and robust scene representation that can benefit various embodied AI tasks.

## ETHICS STATEMENT

The authors have adhered to the ICLR Code of Ethics. Our research utilizes publicly available datasets (TUM-RGBD and ScanNet) and does not involve human subjects or the release of new sensitive data. The authors declare no competing interests and are not aware of any other direct ethical issues arising from this work.

## REPRODUCIBILITY STATEMENT

We are committed to ensuring the full reproducibility of our research. Upon acceptance, we will release our complete source code, which includes all scripts for data preprocessing, the full pipeline implementation, and evaluation. In the interim, we provide extensive details to support this goal. Our methodology and implementation details are thoroughly described in Sec. 4.1 and Appendix A.4. Furthermore, we use publicly available datasets (TUM-RGBD and ScanNet), and our preprocessing is standard and minimal, limited to conventional lens distortion correction and a simple crop to remove peripheral black borders. We are confident that these materials provide a clear basis for the verification and reproduction of our work.

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

# A  APPENDIX

The appendix provides additional experimental results and implementation details.

## A.1  PER-SCENE RENDERING RESULTS

We test our rendering performance on 6 scenes from ScanNet and 3 scenes from TUM-RGBD, conducting a scene-by-scene quantitative comparison with other RGB-D and RGB baselines. The results are shown in Table 6 and Table 7, respectively. Our model achieve the highest scores in PSNR, SSIM, and LPIPS on both ScanNet and TUM-RGBD sequences.

Table 6: **Per-scene rendering** results on ScanNet dataset.

| Method | Metric | 0000 | 0059 | 0106 | 0169 | 0181 | 0207 | Avg. |
|---|---|---|---|---|---|---|---|---|
| *RGB-D* | | | | | | | | |
| SplaTAM | PSNR↑ | 18.70 | 20.91 | 19.84 | 22.16 | 22.01 | 18.90 | 20.42 |
| | SSIM↑ | 0.71 | 0.79 | 0.81 | 0.78 | 0.82 | 0.75 | 0.78 |
| | LPIPS↓ | 0.48 | 0.32 | 0.32 | 0.34 | 0.42 | 0.41 | 0.38 |
| LoopSplat | PSNR↑ | 25.19 | 23.21 | 23.29 | 26.86 | 23.78 | 25.28 | 24.60 |
| | SSIM↑ | 0.85 | 0.83 | 0.84 | 0.88 | 0.80 | 0.83 | 0.84 |
| | LPIPS↓ | 0.44 | 0.40 | 0.41 | 0.34 | 0.53 | 0.44 | 0.43 |
| *RGB* | | | | | | | | |
| MonoGS | PSNR↑ | 16.91 | 19.15 | 18.57 | 20.21 | 19.51 | 18.37 | 18.79 |
| | SSIM↑ | 0.62 | 0.69 | 0.74 | 0.74 | 0.75 | 0.70 | 0.71 |
| | LPIPS↓ | 0.70 | 0.51 | 0.55 | 0.54 | 0.63 | 0.58 | 0.59 |
| GLORIE-SLAM | PSNR↑ | 23.42 | 20.66 | 20.41 | 25.23 | 21.28 | 23.68 | 22.45 |
| | SSIM↑ | 0.87 | 0.83 | 0.84 | 0.91 | 0.76 | 0.85 | 0.84 |
| | LPIPS↓ | 0.26 | 0.31 | 0.31 | 0.21 | 0.44 | 0.29 | 0.36 |
| **EmbodiedGS** | PSNR↑ | 26.13 | 24.10 | 24.59 | 30.88 | 25.87 | 27.31 | 26.48 |
| | SSIM↑ | 0.92 | 0.89 | 0.90 | 0.95 | 0.90 | 0.92 | 0.91 |
| | LPIPS↓ | 0.21 | 0.22 | 0.22 | 0.13 | 0.25 | 0.20 | 0.21 |

Table 7: **Per-scene rendering** results on TUM-RGBD dataset.

| Method | Metric | fr1/desk | fr2/xyz | fr3/off. | Avg. |
|---|---|---|---|---|---|
| *RGB-D* | | | | | |
| SplaTAM | PSNR↑ | 22.00 | 24.50 | 21.90 | 22.80 |
| | SSIM↑ | 0.86 | 0.95 | 0.88 | 0.89 |
| | LPIPS↓ | 0.23 | 0.10 | 0.20 | 0.18 |
| LoopSplat | PSNR↑ | 22.15 | 23.81 | 23.47 | 23.14 |
| | SSIM↑ | 0.85 | 0.92 | 0.88 | 0.88 |
| | LPIPS↓ | 0.31 | 0.19 | 0.25 | 0.25 |
| *RGB* | | | | | |
| MonoGS | PSNR↑ | 19.67 | 16.17 | 20.63 | 18.82 |
| | SSIM↑ | 0.73 | 0.72 | 0.77 | 0.74 |
| | LPIPS↓ | 0.33 | 0.31 | 0.34 | 0.33 |
| GLORIE-SLAM | PSNR↑ | 20.26 | 25.62 | 21.21 | 22.36 |
| | SSIM↑ | 0.87 | 0.96 | 0.84 | 0.89 |
| | LPIPS↓ | 0.31 | 0.09 | 0.32 | 0.24 |
| **EmbodiedGS** | PSNR↑ | 22.78 | 27.10 | 23.53 | 24.47 |
| | SSIM↑ | 0.87 | 0.96 | 0.89 | 0.91 |
| | LPIPS↓ | 0.23 | 0.07 | 0.19 | 0.16 |

## A.2 ADDITIONAL VISUALIZATION

We provide a qualitative evaluation of instance segmentation performance on the ScanNet dataset (Dai et al., 2017) in Figure 6. As illustrated, baseline methods like SAM3D are highly susceptible to noise in the input geometry from MASt3R-SLAM, resulting in fragmented and erroneous instance masks. In contrast, EmbodiedGS exhibits significant robustness to these geometric imperfections, yielding cleaner and more coherent instance segmentations. This underscores our method's ability to jointly refine geometry and perception for superior accuracy.

| Input Scene | Ground Truth | SAM3D‡ | Ours |
|---|---|---|---|

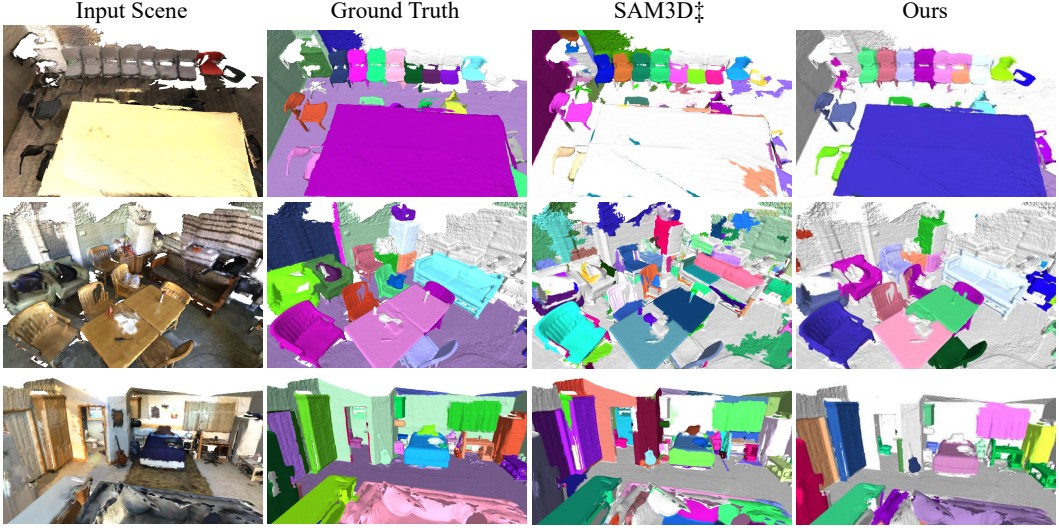

Figure 6: Visualization of instance segmentation results on ScanNet.

We also provide some rendering visualization on both ScanNet and TUM-RGBD (Sturm et al., 2012) datasets. As shown in Figure 7, our approach demonstrates superior rendering quality. In comparison, both GLORIE-SLAM and SplaTAM produce results with significant blurring and artifacts, while LoopSplat struggles with rendering sharp object boundaries.

## A.3 MEMORY FOOTPRINT

We showcase our method's ability to handle long video sequences using Scene0000 in ScanNet dataset (5578 frames). As depicted in Figure 8, by promptly saving and resetting the active submap, we ensure its memory consumption remains manageable throughout the entire process.

| GLORIE-SLAM | SplaTAM | LoopSplat | Ours | Ground Truth |
|---|---|---|---|---|

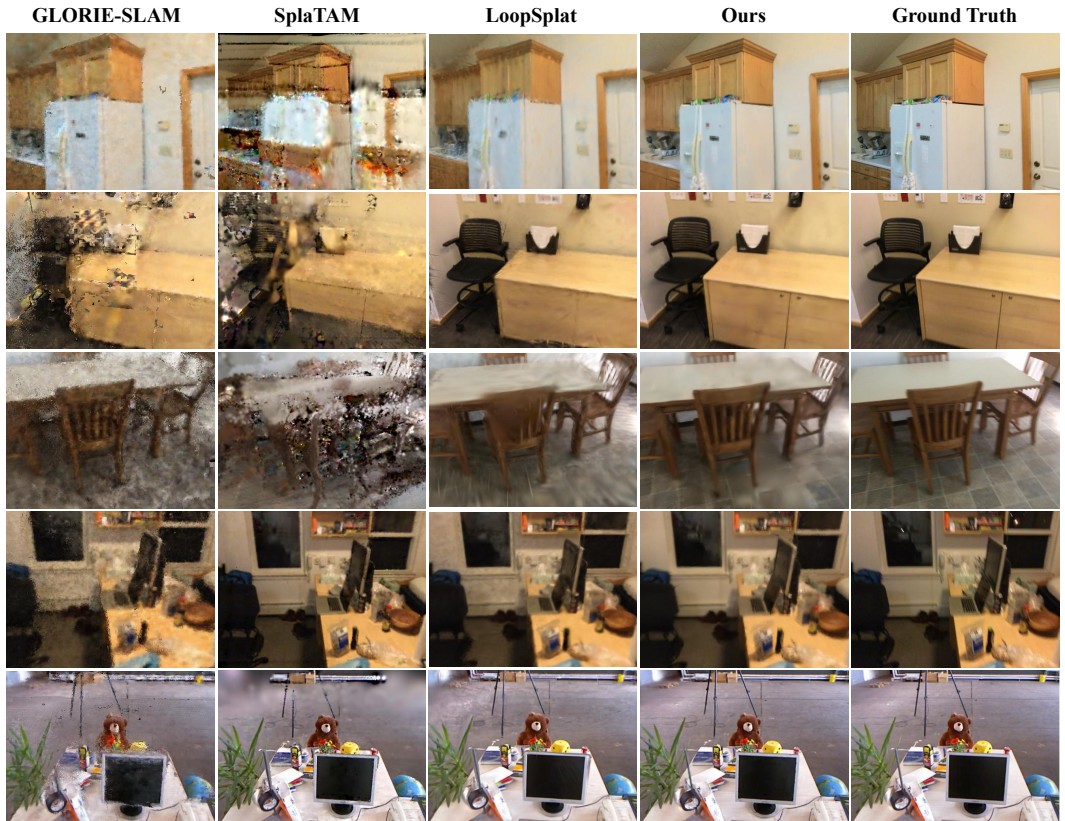

Figure 7: Rendering Visualization on ScanNet and TUM-RGBD datasets.

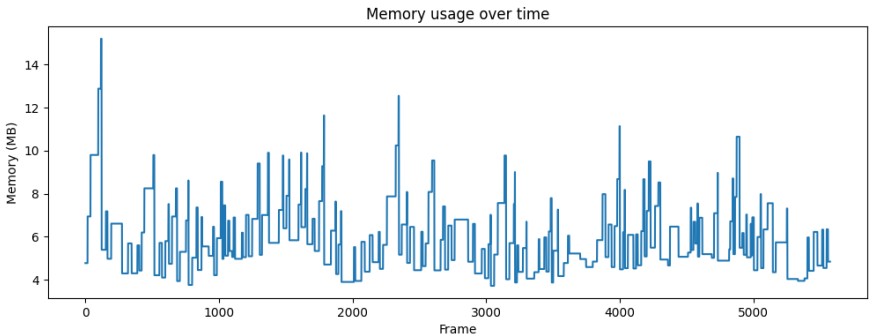

Figure 8: Memory Footprint on ScanNet Scene0000.

## A.4 IMPLEMENTATION DETAILS

**Mapping Frames Selection**.We select mapping frames through the following approach. First, all keyframes are selected as mapping frames to obtain the initial gaussian mapping of the current submap. Additionally, after acquiring the camera pose of the current frame through tracking, we render the gaussians of current submap $\mathcal{G}_{t-1}^{pre}$ onto the current frame to obtain the accumulated opacity map. Areas where accumulated opacity falls below a threshold $\sigma_o$ are considered insufficiently observed. If the proportion of insufficiently observed regions on the current frame exceeds a threshold $\sigma_{vis}$, the current frame is selected as a mapping frame, and the pixels corresponding to insufficiently observed areas are used to initialize new gaussians, which are then added to the current submap. We set $\sigma_o = 0.6, \sigma_{vis} = 0.3$.

**Instance Masks Pre-rendering**. During the rendering of the submap's gaussians on the current frame to compute the accumulated opacity map, we also obtain the corresponding instance embed-

ding map for the current frame. Additionally, we simultaneously render an accumulated instance opacity map by assigning an opacity of 1 to foreground gaussians and 0 to background gaussians. The resulting instance embedding map is explicitly converted into instance masks, retaining only regions where the accumulated instance opacity exceeds a predefined threshold $\sigma_{ins\_o}$ to ensure the precision of the pre-rendered masks. We set $\sigma_{ins\_o} = 0.9$. A visualization of the pre-rendered masks is provided in Figure 9.

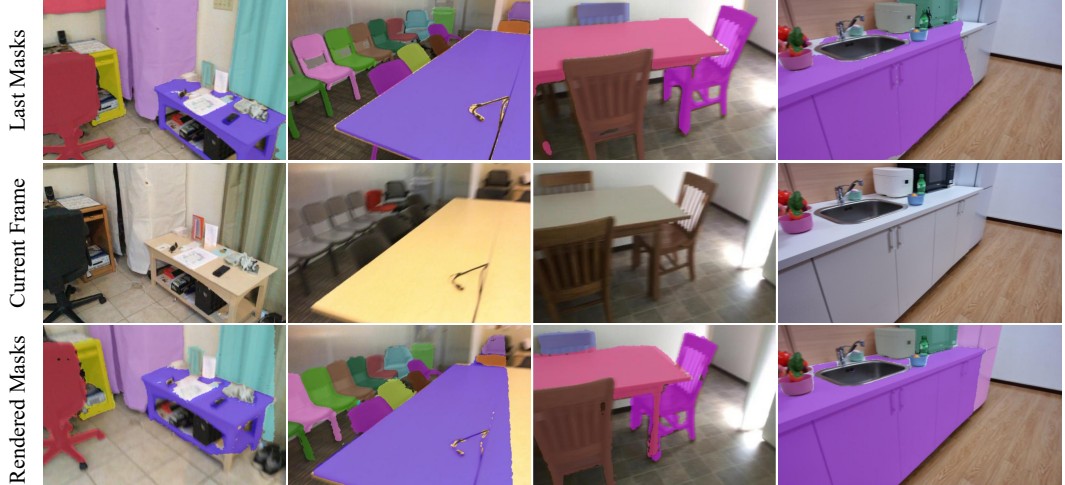

Figure 9: Visualization of Pre-rendered Instance Masks.

**Final Global Instance Merging**. Due to the error accumulation during the tracking process that cannot be entirely eliminated by loop closure optimization, the point cloud position of an instance when it reappears after a long period may differ from its previous position. This could result in a low geometric similarity and lead to matching failures. To compensate for this error, after completing the processing of the entire sequence, we once again extract the 3D bounding box and MASt3R Encoder Features for each instance and calculate the box coverage and feature cosine similarity between every pair of instances. Additionally, during the sequence processing, we also record whether two instances have appeared together in the same frame. Finally, two instances are merged if their box coverage and feature cosine similarity exceed thresholds $\sigma_{box}$ and $\sigma_{feat}$, respectively, and they have never appeared together in the same frame. We set $\sigma_{box} = \sigma_{feat} = 0.5$.

**Final Submaps Merging and Refinement.** After processing the entire sequence, our method merges all submaps to construct a global OCGS representation. To ensure the compactness of this representation and mitigate local overfitting, we subsequently refine it through downsampling and iterative rendering optimization. Specifically, we first aggregate all Gaussians from the submaps, and then apply a downsampling procedure using a voxel grid with a size of 2cm, where only the Gaussian with the highest confidence is retained within each cell. Consequently, we perform global instance merging as is described above. Finally, the global OCGS representation is optimized for $N_{it}$ iterations using the photometric loss to yield the final, refined model. We set $N_{it} = 10000$.

## A.5 MORE DISCUSSIONS

**Scalability.** The EmbodiedGS pipeline is architected for scalability across long sequences, large-scale environments, and numerous object instances. (1) To handle long-duration operations, our submap-based framework dynamically partitions the input stream, ensuring a manageable and consistent memory footprint and preventing unbounded growth, as empirically demonstrated in Figure 8. (2) Our foundation on MASt3R-SLAM provides robustness against pose drift through loop closure, while our Gaussian optimization and P4R loss further improve reconstruction fidelity over time. (3) The binary instance embedding is highly compact, as its required dimension grows only logarithmically with the number of objects—doubling the instance count requires only one additional bit, imposing negligible overhead on efficiency and performance. The capability of our method to handle a large-scale outdoor environment is qualitatively demonstrated in Figure 4.

**Generalization Ability.** Our method's strong generalization capabilities are founded on the strategic integration of powerful, pre-trained Vision Foundation Models (VFMs). In reconstruction, our geometric backbone, MASt3R, is pre-trained on diverse datasets spanning a wide range of indoor and outdoor environments. This endows our pipeline with strong generalization for geometric initialization, allowing it to establish a robust coarse structure for various scene types, which is subsequently refined via Gaussian optimization. In perception, we use the open-vocabulary detector YOLO-E, whose strong performance on both common and long-tail categories allows our system to generalize to various object classes via text prompts. Our method fully leverage these powerful, generalizable priors and then mutually enhance the reconstruction and perception quality through our joint optimization framework, creating a system that is effective across diverse scenes and objects.

**Limitation.** While EmbodiedGS demonstrates robust performance, we acknowledge several limitations. As a modular system, its performance is inherently dependent on its upstream components, MASt3R-SLAM and YOLO-E. Although our joint optimization framework can refine initial inaccuracies, this refinement-based approach has limited corrective capacity against significant foundational errors. Furthermore, while the method remains robust to drastic viewpoint changes thanks to the MASt3R-SLAM and our multi-cue matching, such scenarios can require a denser selection of mapping frames, which in turn impacts computational efficiency. Future work could explore end-to-end trainable paradigms to address these challenges, potentially enhancing both robustness to severe errors and overall efficiency.

## A.6 Algorithm Pseudocode & Notation Tables

We outline the overall pipeline of our method in Algorithm 1. For a clearer exposition of its two core branches, we provide detailed pseudocode in Algorithm 2 and Algorithm 3, respectively.

---

**Algorithm 1** EmbodiedGS: Main Pipeline

---

**Input:** RGB video stream $\mathcal{V} = \{I_1, I_2, \dots\}$, Camera intrinsics $K$
**Output:** Object-Centric 3D Gaussian (OCGS) representation $\mathcal{G}_{final}$
1: Initialize Global-Associated Instance Memory (GAIM) $O \leftarrow \emptyset$
2: Initialize submap collection $\mathcal{S} \leftarrow \emptyset$ and mapping frame collection $\mathcal{V}^{map} \leftarrow \emptyset$
3: Initialize current submap $\mathcal{G}_0^{sub} \leftarrow \emptyset$ and its mapping frames $\mathcal{V}_0^{map} \leftarrow \emptyset$
4: Initialize MASt3R-SLAM tracker and YOLO-E segementer
5: **for** each frame $I_t$ in $\mathcal{V}$ **do**
6: $\qquad\qquad\qquad\qquad\qquad\qquad\qquad\qquad\qquad\qquad\qquad\qquad\qquad$ ▷ — Tracking —
7: $\quad (\mathbf{T}_{kt}, \mathbf{X}_t^t, \mathbf{C}_t^t) \leftarrow$ MASt3R_SLAM.Track$(I_t)$ $\qquad$ ▷ Pose, Pointmap, Confidence
8: $\quad$ **if** $I_t$ is a new mapping frame **then**
9: $\qquad\qquad\qquad\qquad\qquad\qquad\qquad\qquad\qquad$ ▷ — Global Instance Association —
10: $\qquad P_t \leftarrow$ YOLO-E.Segment$(I_t)$ $\qquad\qquad\qquad\qquad$ ▷ Get 2D instance proposals
11: $\qquad$ Extract geometric & visual descriptors for $P_t$
12: $\qquad (O, \mathrm{ID}_t) \leftarrow$ UpdateGAIM$(O, P_t, \mathcal{G}_{t-1}^{sub}, \mathbf{T}_{kt})$ $\qquad\qquad$ ▷ See Alg. 2
13: $\qquad\qquad\qquad\qquad\qquad\qquad\qquad\qquad$ ▷ — Incremental Gaussian Mapping —
14: $\qquad \mathcal{G}_t^{cur} \leftarrow$ InitializeGaussians$(\mathbf{X}_t^t, \mathbf{T}_{kt}, \mathrm{ID}_t)$ $\quad$ ▷ Init. from current pointmap
15: $\qquad$ **if** $I_t$ is a new keyframe **then** $\qquad\qquad\qquad\qquad$ ▷ Submap Management
16: $\qquad\qquad$ Finalize and store current submap: $\mathcal{S} \leftarrow \mathcal{S} \cup \{\mathcal{G}_{t-1}^{sub}\}$
17: $\qquad\qquad$ Store current mapping frames: $\mathcal{V}^{map} \leftarrow \mathcal{V}^{map} \cup \mathcal{V}_t^{map}$
18: $\qquad\qquad$ Initialize new submap $\mathcal{G}_t^{sub} \leftarrow \mathcal{G}_t^{cur}, \mathcal{V}_t^{map} \leftarrow I_t$
19: $\qquad$ **else**
20: $\qquad\qquad$ Merge Gaussians of current frame into submap $\mathcal{G}_t^{sub} \leftarrow \mathcal{G}_{t-1}^{sub} \cup \mathcal{G}_t^{cur}$
21: $\qquad\qquad$ Add current frame to submap mapping frames $\mathcal{V}_t^{map} \leftarrow \mathcal{V}_{t-1}^{map} \cup I_t$
22: $\qquad\qquad\qquad\qquad\qquad\qquad\qquad\qquad\qquad\qquad$ ▷ — OCGS Refinement —
23: $\qquad \mathcal{G}_t^{sub} \leftarrow$ JointOptimization$(\mathcal{G}_t^{sub}, \mathcal{V}_t^{map}, O)$ $\qquad\qquad$ ▷ See Alg. 3
24: $\mathcal{G}_{final} \leftarrow$ MergeSubmaps&Refine$(\mathcal{S}, \mathcal{V}^{map})$
25: **return** $\mathcal{G}_{final}$

---

---

**Algorithm 2** UpdateGAIM

---

**Input:** GAIM $O = \{o_i\}_{i=1}^N$, Current proposals $P_t = \{p_j\}_{j=1}^L$, Submap Gaussians $\mathcal{G}_{t-1}^{sub}$, Pose $\mathbf{T}_t$
**Output:** Updated GAIM, Globally-consistent IDs for proposals $P_t$

1: Initialize similarity matrix $S \in \mathbb{R}^{N \times L}$
2: **for** each instance $o_i \in O$ **do**
3:      **for** each proposal $p_j \in P_t$ **do**
4:                                    ▷ — Compute Multi-Cue Similarities —
5:          $s_{ij}^{geo} \leftarrow \texttt{DIoU}(o_i.\text{box}, p_j.\text{box})$
6:          $s_{ij}^{vis} \leftarrow \texttt{CosineSimilarity}(o_i.\text{vis}, p_j.\text{vis})$
7:          $M_i^{rendered} \leftarrow \texttt{RenderInstanceMask}(\mathcal{G}_{t-1}^{sub}, o_i.\text{id}, \mathbf{T}_t)$
8:          $s_{ij}^{tem} \leftarrow \texttt{CoCoverage}(M_i^{rendered}, p_j.\text{mask})$
9:          $s_{ij} \leftarrow \lambda_1 s_{ij}^{geo} + \lambda_2 s_{ij}^{vis} + \lambda_3 s_{ij}^{tem}$             ▷ Weighted sum
10:          $S[i, j] \leftarrow s_{ij}$
11: matches $\leftarrow \texttt{BipartiteMatching}(S, \text{threshold} = \sigma_m)$
12: **for** each proposal $p_j \in P_t$ **do**
13:      **if** $p_j$ is matched to $o_i$ **then**
14:          Merge $p_j$ into $o_i$ in GAIM (update masks, box, vis. feature)
15:          Assign ID of $o_i$ to $p_j$
16:      **else**
17:          Register $p_j$ as new instance in GAIM w/ new global ID
18:          Assign the new ID to $p_j$
19: **return** (GAIM, Assigned IDs)

---

---

**Algorithm 3** JointOptimization

---

**Input:** Submap Gaussians $\mathcal{G}_t^{sub}$, Mapping frames $\mathcal{V}_t^{map}$, GAIM $O = \{o_i\}$
**Output:** Optimized submap Gaussians $\mathcal{G}_t^{sub}$

1: **for** a fixed number of iterations **do**
2:      $I_m^{map} \leftarrow \texttt{RandomSample}(\mathcal{V}_t^{map})$
3:                                     ▷ — Differentiable Rendering —
4:      $\hat{I}_m^{map} \leftarrow \texttt{RenderColor}(\mathcal{G}_t^{sub}, \mathbf{T}_m)$
5:      $E_m \leftarrow \texttt{RenderEmbedding}(\mathcal{G}_t^{sub}, \mathbf{T}_m)$
6:                                        ▷ — Photometric Loss —
7:      $\mathcal{L}_{pho} \leftarrow (1 - \lambda_{SSIM})\|I_m^{map} - \hat{I}_m^{map}\|_1$
               $+ \lambda_{SSIM}(1 - \text{SSIM}(I_m^{map}, \hat{I}_m^{map}))$
8:                ▷ — R4P: Reconstruction for Perception Loss —
9:      $\tilde{M}_m \leftarrow \texttt{GetMasksFromGAIM}(I_m^{map})$
10:      $\mathcal{L}_{R4P} \leftarrow \texttt{ComputeBCELoss}(\sigma(E_m), \text{binary}(\tilde{M}_m))$     ▷ Weighted by confidences
11:                ▷ — P4R: Perception for Reconstruction Loss —
12:      $\mathcal{L}_{P4R} \leftarrow \texttt{ComputeSmoothnessLoss}(\mathcal{G}_t^{sub})$
13:      $\mathcal{L}_{total} \leftarrow \mathcal{L}_{pho} + \lambda_{R4P}\mathcal{L}_{R4P} + \lambda_{P4R}\mathcal{L}_{P4R}$
14:                                            ▷ — Update —
15:      Update params of all $g \in \mathcal{G}_t^{sub}$ via gradient descent on $\mathcal{L}_{total}$
16: **return** $\mathcal{G}^{sub}$

---

Table 8: Table of Notations

| Symbol | Description |
|---|---|
| **General Symbols & Inputs** | |
| $\mathcal{V}$ | The input RGB video stream, $\{I_1, I_2, \dots\}$. |
| $I_t$ | The RGB image at time $t$. |
| $K$ | Camera intrinsic parameters. |
| $\mathbf{T}_{kt}$ | Relative camera pose (Sim(3) transformation) from keyframe $k$ to current frame $t$. |
| $\mathbf{X}_t^t$ | Pointmap (3D points in camera coordinates) generated for frame $I_t$. |
| $\mathbf{C}_t^t$ | Confidence map associated with the pointmap $X_t^t$. |
| $O$ | The Global-Associated Instance Memory (GAIM), a set of observed instances $\{o_i\}$. |
| $o_i$ | A specific instance stored in GAIM, containing attributes like box, visual feature, etc. |
| $P_t$ | A set of 2D instance proposals $\{p_j\}$ detected in frame $I_t$. |
| $p_j$ | A specific instance proposal, containing mask, box, and visual feature. |
| $\mathrm{ID}_t$ | The set of globally-consistent instance IDs assigned to proposals in $P_t$. |
| $\mathbf{g}$ | A 3D Gaussian point, $\mathbf{g} = (\boldsymbol{\mu}, \mathbf{s}, \mathbf{r}, \alpha, \mathbf{c}, \mathbf{e})$. |
| $\mathcal{G}$ | A general set of 3D Gaussians. |
| $\mathcal{G}_{final}$ | The final, merged OCGS representation of the entire scene. |
| $\mathcal{S}$ | A collection of finalized submaps. |
| $\mathcal{G}_t^{sub}$ | The set of 3D Gaussians in the current submap being actively mapped. |
| $\mathcal{G}_t^{cur}$ | The set of new 3D Gaussians initialized from frame $I_t$. |
| $\mathcal{V}^{map}$ | A collection of all mapping frames from finalized submaps. |
| $\mathcal{V}_t^{map}$ | The set of mapping frames corresponding to the active submap $\mathcal{G}_t^{sub}$. |
| $\lambda, \sigma$ | Hyperparameters (weights, thresholds). |
| **Algorithm 2: GAIM Update** | |
| $S$ | The $N \times L$ similarity matrix between $N$ existing instances and $L$ new proposals. |
| $s_{ij}^{geo}$ | Geometric similarity score (from DIoU) between instance $o_i$ and proposal $p_j$. |
| $s_{ij}^{vis}$ | Visual similarity score (from Cosine Similarity) between $o_i$ and $p_j$. |
| $s_{ij}^{tem}$ | Temporal similarity score (from Co-coverage) between $o_i$ and $p_j$. |
| $M_i^{rendered}$ | The instance mask for instance $o_i$ rendered from the 3D submap. |
| **Algorithm 3: Joint Optimization** | |
| $\hat{I}_m^{map}$ | The color image rendered from the submap $\mathcal{G}^{sub}$. |
| $E_m$ | The instance embedding map rendered from the submap $\mathcal{G}^{sub}$. |
| $\tilde{M}_m$ | The global-associated instance masks for frame $I_m^{map}$, retrieved from GAIM. |
| $\mathcal{L}_{pho}$ | The photometric loss (weighted-sum of L1 and SSIM Loss). |
| $\mathcal{L}_{R4P}$ | The "Reconstruction for Perception" loss (BCE loss on instance embeddings). |
| $\mathcal{L}_{P4R}$ | The "Perception for Reconstruction" loss (intra-instance smoothness). |
| $\mathcal{L}_{total}$ | The final combined loss function for joint optimization. |

### A.7 THE USE OF LARGE LANGUAGE MODELS

Throughout the preparation of this manuscript, we employed a Large Language Model for linguistic refinement. The model was used for improving the conciseness and flow of our sentences, clarifying our technical expressions, and reinforcing the logical structure of our arguments. Apart from these, all core scientific contributions—including the conceptualization of research ideas, methodological design, experimental execution, and the analysis and interpretation of results—were performed by the human authors. All authors take full responsibility for the content and claims of this work.

