# OpenReview forum: "EmbodiedGS: Reconstruct Unified Embodied Representation from RGB Stream"
_ICLR.cc/2026/Conference — ICLR 2026 Conference Withdrawn Submission_

### Official Review · Reviewer_bCqj · 2025-10-28

**Soundness:** 2
**Presentation:** 3
**Contribution:** 2
**Rating:** 2
**Confidence:** 4

**Summary:**

The paper introduces EmbodiedGS, an object-centric online reconstruction and perception framework based on RGB video inputs. The methods combine SOTA SLAM with a new instance association mechanism and propose a novel reconstruction-perception joint optimization process to enhance the geometry and perception results. Extensive experimental results demonstrate the superiority of the proposed EmbodiedGS.

**Strengths:**

The authors introduce a dedicated object-centric Gaussian Representation and reconstruction-perception joint optimization process strategy to improve reconstruction and perception performance in various scenes. The designs are well-supported by experimental results. The methods used are appropriate for the task, and the authors provide a thorough evaluation of the effectiveness of the proposed method.

**Weaknesses:**

1.What is the difference between the setting of the proposed method and semantic SLAM, just use or not use depth? The experimental setting is also same with semantic slam, this is no need to re-name this task?

2.The authors do not provide a thorough evaluation of the effectiveness of the proposed method. What are the instance association mechanism and bidirectional optimization spereate influence on perception and reconstruction. Additionally, there is no clear results analysis on how bidirectional optimization improve perception and reconstruction.

3.The novelty of instance association mechanism is limited and there are lots of hyper-parameters to be tuned, resulting in instable performance in different scenes.

4.The explanation of bidirectional optimization is unclear. How exactly reconstruction and perception optimize each other?

**Questions:**

1.It seems that instance embedding is not used after optimized with Eq. 7, how the instance embedding affect the reconstruction and perception results?

2.How the \hat{M_m(p)} is obtained, how b() maps a mask to d-bit binary encoding?

3.How Eq.7 and Eq.8 optimize each other? It seems they all try to make instance embedding/Gaussian align to 3d mask, reducing intra-object spatial variance.

4.There is no embodied related experiments such as navigation, manipulation, the title “EmbodiedGS” is not suitable for this paper.

---

### Official Review · Reviewer_Kr1V · 2025-10-30

**Soundness:** 3
**Presentation:** 3
**Contribution:** 1
**Rating:** 2
**Confidence:** 5

**Summary:**

The proposed method reconstructs object segmentation from an unposed RGB stream. Given an RGB video, it employs MASt3R-SLAM to compute camera poses and depth maps, and utilizes YOLO-E for segmentation prediction. The segmentation results are then projected onto the depth maps and used to train a Gaussian Splatting model. The visualizations demonstrate promising results.

**Strengths:**

1. The writing is easy to follow.
2. The visualization looks good.

**Weaknesses:**

1. The computation of the bounding box is not mentioned in the paper.
2. In Line 253, although the ablation study shows that visual similarity contributes to the result to some extent, there are no explanations or experiments to validate the multi-view feature descriptor strategy. Previous literature indicates that feature descriptors can vary significantly across different views.
3. In Line 279, the term "Binary Instance Embedding" (BIE) is questionable. It is unsurprising that BIE works in a submap Gaussian Splatting system, as it appears to overfit, similar to an image representation, and fails in global coordinates. The primary reason is that Gaussian representations are sensitive and prone to overfitting the given observations. Even if we render RGB images from the Gaussian Splatting system from limited views, it may not generalize well to novel views with slightly larger baselines. Given the inherent discreteness of binary representations, I have strong doubts about the performance of Equation 6. Additionally, upon reviewing the figures, the authors only provide visualizations of instance IDs on point clouds.
4. The overall technical contribution of this paper is limited. It seems more suitable for a robotics conference rather than ICLR, because there are tons of paper with similar instance fusing strategies published on iros or icra.
5. The authors report numerous results on rendering quality and runtime. However, the PSNR and runtime metrics are fully provided by the Mast3R-SLAM framework, and based on Equations 7 and 8, the author's contribution to these aspects appears minimal.
9. Some baselines are missing, for example, BoxFusion [1].

[1] BoxFusion: Reconstruction-Free Open-Vocabulary 3D Object Detection via Real-Time Multi-View Box Fusion

**Questions:**

1. Based on the paper, the yolo-e only used to extract the mask(line180). Why the visual descriptor is extracted from mast3r but not yoloe?

---

### Official Review · Reviewer_Gje1 · 2025-11-01

**Soundness:** 3
**Presentation:** 3
**Contribution:** 3
**Rating:** 8
**Confidence:** 5

**Summary:**

In this paper authors presented an efficient pipeline for online reconstruction from RGB video streams. Sufficient experiments are shown  on real-world datasets demonstrate the proposed method effectively constructs high-fidelity 3D, achieving superior performance in both segmentation and rendering quality.

**Strengths:**

The use of Global-Associated Instance Memory (GAIM) by incrementally associating instance proposals across views with multi-modal matching criteria to ensure global consistency is the key highlight. Additionally key strength is in joint optimization approach of the OCGS by leveraging the synergy between perception and reconstruction, and simultaneously refining Gaussian geometry and instance embedding to produce a high-fidelity representation.

**Weaknesses:**

- experimental could have been better. Currently, all scenes considered are static, what about those scenarios where there are moving objects.
- How varying light conditions and texture vs texture less environment will affect the overall performance of the system.
- An analysis on time taken for inference is missing. A section talking about time profiling will help understand the utility of this system for real world applications

**Questions:**

Please see above

---

### Official Review · Reviewer_TtH8 · 2025-11-01

**Soundness:** 2
**Presentation:** 3
**Contribution:** 2
**Rating:** 6
**Confidence:** 4

**Summary:**

This paper presents a pipeline for incremental object-centric 3D Gaussian reconstruction from monocular RGB streams. Specifically, it combines MASt3R-SLAM for initialization with a GAIM for cross-view tracking, and introduces a joint optimization that refines OCGS. The experiments validate its effectiveness on several datasets.

**Strengths:**

* The pipeline integrates geometry, instance tracking, and rendering within a unified Gaussian-splatting framework.
* The proposed pipeline is validated on TUM-RGBD and ScanNet, as well as real-world demonstrations.
* The bidirectional optimization (R4P/P4R) between reconstruction and perception is conceptually interesting.
* The paper is clear in general and easy to follow.

**Weaknesses:**

* The method essentially replaces the geometric backbone of prior object-centric SLAM systems with MASt3R-SLAM and integrates standard 2D object tracking (YOLO-E + mask association). Similar systems have appeared in semantic/instance-aware SLAM  works (e.g., SGS-SLAM, SemGauss-SLAM). Acutally, the idea of utilizing 2D segmentation to improve VO/SLAM is quite old and standard in both CV and robotics communities, like VSO [Lianos et al. 2018] and its following works.
* The framework relies heavily on 2D instance segmentation quality. The robustness of the system depends on YOLO-E’s accuracy. In cluttered or textureless scenes with few distinct objects, GAIM would likely fail, and the pipeline may degrade to vanilla MASt3R-SLAM performance.
* Since MASt3R-SLAM itself still suffers from pose drift and unstable geometry, errors will propagate through the entire system. The “joint optimization” seems unable to fully correct such foundational errors in principle since it also highly depends on the initialization.
* The multi-branch architecture (GAIM + submap + joint optimization) substantially increases system complexity and computational overhead. The performance gain may not justify the added complexity for practical deployment. Actually, as for the real application for robotics, the system efficiency is of great importance.
* Table 5 and some other ablations do not specify on which dataset or sequence they were performed. Moreover, the reported improvements are marginal.

**Questions:**

Please refer to the weaknesses in the weaknesses part. I am open to being persuaded based on the feedback from the authors.

---

### Note · Authors · 2025-11-14

I have read and agree with the venue's withdrawal policy on behalf of myself and my co-authors.